# Automatic Modulation Classification for Underwater Acoustic Communication Signals Based on Deep Complex Networks

**DOI:** 10.3390/e25020318

**Published:** 2023-02-09

**Authors:** Xiaohui Yao, Honghui Yang, Meiping Sheng

**Affiliations:** School of Marine Science and Technology, Northwestern Polytechnical University, Xi’an 710060, China

**Keywords:** automatic modulation classification, underwater communication signals, underwater acoustic channel, deep complex networks

## Abstract

Automatic modulation classification (AMC) is an important method for monitoring and identifying any underwater communication interference. Since the underwater acoustic communication scenario is full of multi-path fading and ocean ambient noise (OAN), coupled with the application of modern communication technology, which is usually susceptible to environmental influences, automatic modulation classification (AMC) becomes particularly difficult when it comes to an underwater scenario. Motivated by the deep complex networks (DCN), which have an innate ability to process complex data, we explore DCN for AMC of underwater acoustic communication signals. To integrate the signal processing method with deep learning and overcome the influences of underwater acoustic channels, we propose two complex physical signal processing layers based on DCN. The proposed layers include a deep complex matched filter (DCMF) and deep complex channel equalizer (DCCE), which are designed to remove noise and reduce the influence of multi-path fading for the received signals, respectively. Hierarchical DCN is constructed using the proposed method to achieve better performance of AMC. The influence of the real-world underwater acoustic communication scenario is taken into account; two underwater acoustic multi-path fading channels are conducted using the real-world ocean observation dataset, white Gaussian noise, and real-world OAN are used as the additive noise, respectively. Contrastive experiments show that the AMC based on DCN can achieve better performance than the traditional deep neural network based on real value (the average accuracy of the DCN is 5.3% higher than real-valued DNN). The proposed method based on DCN can effectively reduce the influence of underwater acoustic channels and improve the AMC performance in different underwater acoustic channels. The performance of the proposed method was verified on the real-world dataset. In the underwater acoustic channels, the proposed method outperforms a series of advanced AMC method.

## 1. Introduction

Automatic modulation classification (AMC) plays an important role in underwater acoustic communication, especially in non-cooperative communication. AMC can be used in monitoring and identification of communication interference, such as spectrum surveillance and electronic countermeasure [1]. Modern underwater platforms (such as submarines, autonomous underwater vehicles (AUV), and underwater unmanned vehicles (UUV)) are in urgent need of advanced AMC technology.

AMC is the intermediate process of signal detection and demodulation [2], occupying a very important position in signal processing technologies. Currently, most of the research on AMC focuses on radio signals (e.g., radar, cellular, satellite), and there is relatively little research on the classification of underwater communication signals. However, the demand for AMC in underwater acoustic communication is increasingly urgent with the development of modern communication technology. AMC of underwater acoustic communication signals is an important approach to the acquisition and analysis of target information in marine battlefields and has become a research hotspot [3]. The essential difference between underwater acoustic communication and radio communication is that the latter generally uses electromagnetic waves as the carrier, while the former uses sound waves generally because the conductivity of ocean water makes electromagnetic waves attenuate rapidly, especially at a higher frequency. Although sound wave is more suitable for underwater communication, it is still challenged by the poor underwater acoustic channel. The low velocity of soundwaves in water (about 1500 m/s) results in a significant delay in underwater communication (about 66 ms/km). Moreover, low velocity is one of the fundamental causes of the multi-path effect and Doppler shift [4], coupled with the extensive ocean ambient noise (OAN) [5], distorting the signals in both time and frequency domains, such as time delay-spread, frequency-selective fading, inter-symbol interference (ISI), frequency shift, etc. For all of these reasons, the AMC of underwater acoustic communication signals faces serious challenges.

Current AMC algorithms can be classified into two categories: the maximum likelihood ratio method based on hypothesis testing and the statistical pattern classification method, of which the latter is more commonly used. The pattern classification method relies on two processes. The first is feature extraction, which is widely used in many fields [6,7]. There are a few feature extraction methods used in AMC, including instantaneous statistics features(envelope, frequency, phase, etc.) [8], constellation features [9], spectrum features [3,10], high-order cumulant features (HOC) [11,12], cyclostationary features [13,14], the wavelet features [12,15], etc. The second is classification; widely used classifiers include decision tree classifier [16], neural network classifier [17], support vector machine (SVM) [18], etc. Some methods mentioned above have been applied to the AMC of underwater acoustic communication signals. Wu [14] employed cyclostationary analysis to perform modulation detection on underwater acoustic communication signals, but the proposed method can only identify lower-order phase shift keying (PSK) from higher-order PSK and quadrature amplitude modulation (QAM). Sanderson [13] proposed hierarchical blind AMC schemes for underwater acoustic communication signals, in which second-order cyclostationary features and maximum likelihood were used to identify BPSK, QPSK, and 16QAM. Zhao [19] proposed an AMC method for underwater acoustic communication signals, extracted features of energy entropy of Stockwell-transform spectrogram, and classified them using SVM. Fang [20] proposed a likelihood-based algorithm for identifying different PSK modulations based on sparse Bayesian learning with expectation maximization.

In recent years, deep learning [21] has emerged as a new area of machine learning and has shown remarkable results in computer vision [22], natural language processing [23], and speech recognition [24]. Deep learning techniques allow for building hierarchical representations and learning high-level features from raw data automatically without much prior knowledge. Many deep neural networks (DNN) have been proposed for various tasks, such as deep belief networks (DBN) [21,25], convolution neural networks (CNN) [22], long short-term memory (LSTM) [26], autoencoder (AE) [25,27], VGGnet [28], GoogleNet [29], ResNet [30], generative adversarial networks (GAN) [31], etc. All the above deep learning architectures enable deep learning to excel in many fields, including AMC. Researchers in wireless communication have introduced deep learning into AMC with some success [32,33]. The theoretical advantages of deep learning and its success in many fields have inspired researchers to apply it to the AMC of underwater acoustic communication signals. Yang [32] introduced denoising autoencoder (DAE) into AMC of underwater acoustic communication signals; they considered an underwater multi-path fading channel with additive white Gaussian noise (AWGN). Ding [34] proposed an AMC method that combined CNN and LSTM and achieved the classification of single side band (SSB), frequency modulation (FM), four frequency shift keying (4FSK), orthogonal frequency division multiplexing (OFDM), QPSK and multi-carrier multiple frequency shift keying (MC-MFSK). Yao [1] used GAN to enhance the signals and showed well robustness under different underwater acoustic channels. Deep learning has shown certain advantages in the AMC of underwater acoustic communication signals, but it is still limited by the complexity of modulation types and the poor underwater acoustic channel.

Since deep learning was proposed, almost all architectures, techniques, and blocks of deep learning are based on real value because most computer vision, natural language, and speech datasets are real-valued. However, complex baseband representation is commonly used in communication systems. Most signal processing methods based on deep learning usually treat the real and imaginary parts of complex baseband signals as independent channels, which may result in the merging of in-phase and orthogonal components and loss of the basic polar representation properties [35]. Compared with the real-valued DNN, deep complex networks (DCN) are designed in the rule of complex arithmetic to keep the complex domain features in the network. It is more reasonable and easier to design the signal processing physical layers in DCN, and it is more appropriate to use DCN to process complex baseband signals. On the other hand, complex numbers can have a richer representational capacity theoretically [36,37]. To exploit the advantage of complex-valued representations, Trabelsi et al. [38] provided the key atomic components for DCN, including complex convolution (CConv), complex batch-normalization (CBN), complex weight initialization, and complex activation functions. Using complex values can make contributions to the networks to have better representational capacity in terms of computational and signal processing perspectives.

In this paper, we proposed a new AMC method for AMC of underwater acoustic communication signals based on DCN. Our contributions are as follows:We adopted DCN to AMC of underwater acoustic communication signals to adequately learn features from the raw complex baseband signals.Two physical signal processing layers were constructed based on DCN to improve the AMC performance, including a deep complex matched filter (DCMF) and deep complex channel equalizer (DCCE). The DCMF can help to remove noise from the signals, and the DCCE can reduce the influence of multi-path fading effectively. DCMF and DCCE were embedded in DCN to improve the AMC performance.The influence of underwater acoustic channels on the signals was fully considered, and different underwater acoustic channels were simulated by using the real-world ocean observation dataset and ambient noise. The effectiveness of the proposed method was verified in different underwater acoustic channels and real-world dataset.

This paper is organized as follows. Section 2 presents a review of underwater acoustic communication signals and channels. Then it introduces the proposed AMC method of underwater acoustic communication signals based on DCN. In Section 3, we evaluate the performance of the proposed method with simulation experiments. Finally, the conclusion of the paper is given in Section 4.

## 2. Materials and Methods

### 2.1. Underwater Acoustic Communication Signals and Channel

#### 2.1.1. Signal Model

PSK and QAM have commonly used modulation types in the field of underwater acoustic communication. Both modulation processes of PSK and QAM involve phase keying. PSK and QAM can be formulated, respectively:(1)$$\begin{array}{cc} \; & \mathrm{PSK}:s\left(t\right)=Ae^{j(2\pi f_{c}t+\phi )}\sum_{n=0}^{N}e^{j\theta_{n}}p\left(t-nT_{s}\right) \end{array}$$(2)$$\begin{array}{cc} \; & \mathrm{QAM}:s\left(t\right)=Ae^{j(2\pi f_{c}t+\phi )}\sum_{n=0}^{N}s\left(n\right)e^{j\theta_{n}}p\left(t-nT_{s}\right) \end{array}$$
where *A* is the amplitude of the signal, $f_{c}$ is the carrier frequency offset, ϕ is the phase offset of carrier frequency, *n* is the symbol index, *N* is the total number of symbols, $p\left[t\right]$ is the baseband pulse shaping filter, $T_{s}$ is the symbol period, $e^{j\theta_{n}}$ and ${s\left(n\right)e}^{j\theta_{n}}$ map the transmitted symbols of PSK and QAM, respectively.

#### 2.1.2. Underwater Acoustic Channel

In the real-world underwater acoustic communication scenario, the influence of underwater acoustic channels on communication signals mainly includes two aspects: multi-path fading and OAN, as is shown in Figure 1.

(1) Multi-path fading

Multi-path propagation often exists in underwater acoustic communication. In a multi-path environment, the received signal can be represented as the superposition of a number of time-delayed and amplitude–attenuated versions of the transmitted signal. A typical underwater acoustic channel with multi-path fading can be seen as a filter whose impulse response function is $h\left(t,\tau \right)$, in which τ is the delay time. The impulse response reflects the properties of the multi-path fading channel and can be expressed as
(3)$$h\left(t,\tau \right)=\sum_{k=1}^{K}a_{k}\left(t\right)\delta \left(\tau -\tau_{k}\left(t\right)\right)$$
where *K* is the total number of paths, $\delta \left(\cdot \right)$ is the delta function, $a_{k}$ and $\tau_{k}$ are the attenuation, and the delay of the *k*-th path.

Since the source and receiver are usually not stationary and underwater acoustic reflection boundaries are unstable in most cases, multi-path fading is often accompanied by Doppler shift. We consider Doppler shift caused by relative motion in multi-path propagation. In a multi-path fading channel, each path has an independent Doppler shift factor $f_{dk}$, which can be expressed as
(4)$$f_{dk}=\frac{v_{k}}{c}f_{c}$$
where $f_{c}$ is the carrier frequency of the transmitted signal, and $v_{k}$ is the radial velocity of the source relative to the receiver of the *k*-th path.

Assume the transmitted signal is $s\left(t\right)$, then the received signal $x\left(t\right)$ propagates through underwater acoustic channel can be expressed as
(5)$$x\left(t\right)=\sum_{k=1}^{K}a_{k}\left(t\right)s\left(t-\tau_{k}\left(t\right)\right)e^{j2\pi f_{dk}t}+n\left(t\right)$$
where n(t) is the additive noise of the channel.

(2) Ocean ambient noise

Ocean ambient noise (OAN) is an additive interference in underwater acoustic channels. The composition of the OAN is very complicated and full of impulsive interference due to numerous noise sources, such as ship-radiated noise, industrial noise, wind noise, biological noise, etc. The reasons mentioned above make the OAN cannot be simulated accurately using white Gaussian noise. AMC method based on Gaussianity assumptions will suffer degradation in their performance to a low level. We use real-world OAN as the additive noise of underwater acoustic channels to enhance the robustness of the proposed AMC method in real-world underwater acoustic communication scenarios.

### 2.2. DCN-Based AMC Method

In this section, we presented the proposed AMC method for underwater acoustic communication signals based on DCN. DCMF and DCCE were proposed to improve the performance of DCN for AMC. We used the raw I/Q signals as the input of the DCN without any feature extraction, by which the DCN was forced to learn features from high-dimension complex time sequences. The framework of the proposed AMC method is illustrated in Figure 2.

#### 2.2.1. Deep Complex Matched Filter

The matched filter is often used to improve the signal-to-noise ratio (SNR) of the received signal and optimize the quality of communication, which is an important module in communication. We proposed a deep complex matched filter (DCMF) to improve the SNR of the signals and optimize the classification performance. The DCMF in Figure 2 is made up of a modified CConv layer, which is optimized by an objective function.

The proposed DCMF has three input channels and three output channels. The first input channel of DCMF accepts received signals (x(t)) which are affected by an underwater acoustic channel (multi-path fading and OAN), x(t) propagates through the network, eventually producing categories. The second input is the clean received signals without noise ($x_{c}\left(t\right)$), which is only affected by the multi-path fading. The last input channel accepts additive noise (n(t)). Clean received signals without noise ($x_{c}\left(t\right)$) and the additive noise n(t) are used to optimize the DCMF. The outputs $x_{1}\left(t\right)$, $x_{c1}\left(t\right)$, and $n_{1}\left(t\right)$ are the filtered signals corresponding to the three inputs, respectively. The forward propagation of DCMF can be formulated as
(6)$$\left[\begin{array}{c} x_{1}\left(t\right)\\ x_{c1}\left(t\right)\\ n_{1}\left(t\right) \end{array}\right]=f\left(W_{1}*\left[\begin{array}{c} x(t)\\ x_{c}\left(t\right)\\ n(t) \end{array}\right]\right)+b_{1}$$
where the f(·) is a linear complex activation function. * represents the CConv operation, $W_{1}$ and $b_{1}$ are the complex weight and complex bias of the DCMF. The CConv kernels of the DCMF can be seen as a matched filter bank. In the forward propagation of the DCN, the three inputs of the DCMF are filtered by the filter bank, and the three inputs share the same filters. In the backpropagation, the parameters are updated by the gradient of the objective function, as is shown in Figure 2. The objective function is the SNR increment from input signals to output signals. The DCMF is trained to maximize the objective function. The training process can be formulated as a minimization process:(7)$$\underset{\theta_{dcmf}}{\arg\min}F_{dcmf}=\gamma_{in}-\gamma_{out}$$
where $\theta_{dcmf}$ is the trainable parameters of DCMF, $F_{dcmf}$ is the objective function, $\gamma_{in}$ and $\gamma_{out}$ are the SNR of input and output signals, respectively. Note that each output of DCMF is made up of a group of one-dimensional filtered signals and each of them corresponds to one CConv kernel of DCMF, so the $\gamma_{out}$ is calculated at all dimensions. On the other hand, Equation (Equation 7) can be converted to a more computationally appropriate format:(8)$$\underset{\theta_{dcmf}}{\arg\min}F_{dcmf}=\frac{E{∥x_{c}\left(t\right)∥}^{2}E{∥n_{1}\left(t\right)∥}^{2}}{E{∥x_{c1}\left(t\right)∥}^{2}E{∥n(t)∥}^{2}}$$

#### 2.2.2. Deep Complex Channel Equalizer

The multi-path fading in the underwater acoustic channels will cause ISI, which will degrade the performance of AMC. We propose DCCE to overcome the influence of multi-path fading by reconstructing the transmitted signal s(t) from the signals distorted by the multi-path fading. The DCCE is designed to accept the outputs of DCMF, and there are two input and output channels of DCCE. The first input channel accepts $x_{1}\left(t\right)$, while the second accepts the sum of $x_{c1}\left(t\right)$ and $n_{1}\left(t\right)$. The outputs $x_{2}\left(t\right)$ and $x_{n2}\left(t\right)$ are the equalized signals corresponding to the inputs, respectively. There are several kernel boxes in DCCE, which are independent of each other. Each box accepts one feature map of the input data, which means that the number of boxes are equal to the number of kernels in DCMF. Two CConv layers were designed in one box, and each of them has one complex kernel. The difference between DCCE from ordinary CConv layer is that the kernels of the DCCE are individual, we only perform convolution operations in the time direction of the input signal. Each kernel connects to one output channel of DCMF, as is shown in Figure 2. The processing of DCCE can be formulated as
(9)$$\left[\begin{array}{c} x_{2}^{i}\left(t\right)\\ x_{n2}^{i}\left(t\right) \end{array}\right]=f\left(W_{2b}^{i}*f\left(W_{2a}^{i}*\left[\begin{array}{c} x_{1}^{i}\left(t\right)\\ x_{c1}^{i}\left(t\right)+x_{n1}^{i}\left(t\right) \end{array}\right]\right)+b_{2a}^{i}\right)+b_{2b}^{i},i=1,2,3,\ldots ,M.$$
where *M* is the number of boxes in DCCE, the superscript *i* means the *i*-th feature map of input data or the *i*-th box in DCCE, $W_{2a}^{i}$, $W_{2b}^{i}$, $b_{2a}^{i}$, and $b_{2b}^{i}$ are the complex weight, and complex bias of DCCE in the *i*-th box, f(·) is a linear complex activation function.

The transmitted signals s(t) are processed by an individual CConv layer. The number of kernels of this CConv layer is *M*, and the width of each kernel is 1. This CConv layer was designed to map s(t) several feature maps to match the dimension of $x_{n2}\left(t\right)$, and the randomly initialized complex kernels can ensure that each feature map is not equal to another, including amplitude and phase. The output of this CConv layer is $s_{1}\left(t\right)$. The objective function is the mean square error (MSE) of $x_{n2}\left(t\right)$ and $s_{1}\left(t\right)$:(10)$$\underset{\theta_{dcce}}{\arg\min}F_{dcce}=E{∥s_{1}\left(t\right)-x_{n2}\left(t\right)∥}^{2}$$
where $\theta_{dcce}$ is the trainable parameters of DCCE, $F_{dcce}$ is the objective function, the objective function is optimized to minimize the MSE of $x_{n2}\left(t\right)$ and $s_{1}\left(t\right)$.

#### 2.2.3. Training Method

In the training process, $x_{c}\left(t\right)$, n(t), and s(t) are used to optimize the DCMF and DCCE. The received signals x(t) and the corresponding true labels *y* are used to optimize the rest trainable parameters, which are noted as $\theta_{r}$. The loss function used to optimize $\theta_{r}$ is cross entropy:(11)$$\underset{\theta_{r}}{\arg\min}F_{r}=-\sum_{i=1}^{N_{c}}y^{i}\log\left(y_{p}^{i}\right)$$
where $N_{c}$ is the number of classes, $y_{p}$ is the predicted label, *i* is the *i*-th element of the label. $\theta_{dcmf}$, $\theta_{dcce}$ and $\theta_{r}$ will be updated per epoch. Each of $\theta_{dcmf}$ and $\theta_{dcce}$ will be frozen when the corresponding loss stop decreases. The training will be stopped when the loss of $\theta_{r}$ shows no improvement or the training step reaches the maximum iterations, as is shown in Algorithm 1.
**Algorithm 1** Training of the proposed method.**Input:** received signals x(t), true labels *y*, clean received signals without noise $x_{c}\left(t\right)$, additive noise n(t), transmitted signals s(t), maximum iterations *N*.**Output:** trained parameters of Net: ${\hat{\theta }}_{dcmf}$, ${\hat{\theta }}_{dcce}$, ${\hat{\theta }}_{r}$.**Initialize:** trainable parameters of Net: $\theta_{dcmf}$, $\theta_{dcce}$, $\theta_{r}$, $Update_{dcmf}=True$, $Update_{dcce}=True$.1:**for** i=1,2,…,N**do**2:     $x_{c1}\left(t\right),n_{1}\left(t\right),x_{n1}\left(t\right),s_{1}\left(t\right),x_{n2}\left(t\right),y_{p}=Net\left(x\left(t\right),x_{c}\left(t\right),n\left(t\right)\right)$. ▹ forward propagation, $y_{p}$ is the predicted label.3:     **if** $Update_{dcmf}$ **then**4:          $loss\left(\theta_{dcmf}\right)=F_{dcmf}\left(x_{c}\left(t\right),n\left(t\right),x_{c1}\left(t\right),n_{1}\left(t\right)\right)$5:          **if** $loss(\theta_{dcmf})$ stop decreasing **then**6:               $Update_{dcmf}\leftarrow False$7:          **else**8:               update $\theta_{dcmf}$ to minimize $loss(\theta_{dcmf})$9:          **end if**10:    **end if**11:    **if** $Update_{dcce}$ **then**12:          $loss\left(\theta_{dcce}\right)=F_{dcce}\left(s_{1}\left(t\right),x_{n2}\left(t\right)\right)$13:          **if** $loss(\theta_{dcce})$ stop decreasing **then**14:               $Update_{dcce}\leftarrow False$15:          **else**16:               update $\theta_{dcce}$ to minimize $loss(\theta_{dcce})$17:          **end if**18:    **end if**19:    $loss\left(\theta_{r}\right)=F_{r}\left(y,y_{p}\right)$20:    update $\theta_{r}$ to minimize $loss(\theta_{r})$21:**end for**22:**return**${\hat{\theta }}_{dcmf}$, ${\hat{\theta }}_{dcce}$, ${\hat{\theta }}_{r}$=$\theta_{dcmf}$, $\theta_{dcce}$, $\theta_{r}$.

In the predicting process, only the first input channel of DCMF is opened, and the received signals x(t) come from the testing dataset will be tested. Then the AMC accuracy will be obtained by the predicted labels and the true labels.

## 3. Experiments and Discussion

In this section, extensive simulations were conducted to verify the performance of the proposed AMC method based on DCN:

(1) Experimental dataset under various conditions were built by simulations.

(2) We analyzed the influence of underwater acoustic channels on AMC performance.

(3) We compared the AMC performance between DCN and classical real-valued DNN.

(4) We analyzed the performance of DCMF by visualizing the kernels, then the performance gain of the proposed AMC method was verified by a series of contrastive experiments, including verifying real-world underwater acoustic communication signals.

(5) The AMC performance comparison between the proposed method and some other AMC methods were carried out.

(6) Comparison experiments were carried out to analyze the limitations of the proposed method.

The results in this section were the average values over multiple runs.

### 3.1. Dataset and Parameters

#### 3.1.1. Signals Generation

We considered six commonly used modulation types, including BPSK, QPSK, 8PSK, 16QAM, 32QAM, and 64QAM. The parameters for simulation are presented in Table 1. The SNR ranges from −9 dB to 21 dB with an interval of 3 dB. The received signals were expressed as the sampled complex baseband, the dimension of each signal sample is 6000 × 2, and the duration is 0.5 s. 10,000 signals are generated for each modulation type at each SNR, which contains 5000 training signals, 2000 validation signals, and 3000 testing signals, making a total of 330,000 training signals, 132,000 validation signals, and 198,000 testing signals.

#### 3.1.2. Underwater Acoustic Channel

A real-world underwater acoustic communication scenario was considered in the following experiments. The ocean observation dataset of the chosen area came from the global gridded Argo dataset 2020 (BOA_Argo) [39]. We chose an area of the ocean near the geographic coordinate of 26.5${}^{{}^{\circ}}$ N, 127.5${}^{{}^{\circ}}$ E, as is shown in Figure 3.

The dataset contains temperature, salinity, and static pressure for different months; we selected the data in January. The sound velocity profile can be calculated from temperature, salinity, and pressure by
(12)$$\begin{array}{cc} c= & \;1449.2+4.6T-0.055T^{2}+0.00029T^{3}\\ \; & +(1.34-0.1T)\times (S-35)+0.168P \end{array}$$
where *T* is the temperature, *S* is the salinity, and *P* are the static pressure, which is positively related to the depth. The depth of the chosen area is 460 m, and the sound velocity profile is shown in Figure 4.

As we can see that the sound gradient in this area presents positive and negative at different depths. We put one transmitter ($T_{x}$) and two receivers ($R_{x1}$ and $R_{x2}$) in the underwater channel. The horizontal distance between the transmitter and the two receivers are 3 km and 5 km, respectively. The depths of the transmitter and receivers are 30 m and 80 m, respectively. The transmitter moves towards the receivers at a horizontal speed of 5 m/s, and the receivers keep stationary. Thus, we get two multi-path fading channels (Ch1 and Ch2) with different impulse responses. The time delays and amplitudes of the two channels are illustrated in Table 2, in which the modules of the amplitudes are normalized to [0,1].

We built several underwater acoustic channels based on Ch1 and Ch2 using real-world OAN and AWGN as the additive noise, respectively, in which AWGN is used as a benchmark. The total duration of the noise is 6 h. Figure 5 has shown the spectrogram of one OAN sample.

### 3.2. Experiment Results Analysis

In all classification experiments, we used classification accuracy to evaluate the classification performance. We just calculated the accuracy at each SNR point; the whole symbol rate range and roll-off value range were covered in each classification task. We tested each modulation type at each SNR point; the accuracy of one modulation type at one SNR point is ${Acc}_{1}$. Furthermore, the accuracy of all modulation types at one SNR point (${Acc}_{2}$), and the average accuracy of all modulation types at all SNR points (${Acc}_{3}$) are calculated from ${Acc}_{1}$. The calculation method can be described as
(13)$$\begin{array}{cc} \; & {Acc}_{2}=\frac{1}{N_{m}}\sum_{i=1}^{N_{m}}Acc_{1_{ij}} \end{array}$$
(14)$$\begin{array}{cc} \; & {Acc}_{3}=\frac{1}{N_{snr}N_{m}}\sum_{j=1}^{N_{snr}}\sum_{i=1}^{N_{m}}Acc_{1_{ij}} \end{array}$$
where $N_{m}$ is the number of modulation types, $N_{snr}$ is the number of SNR points.

#### 3.2.1. Influence Analysis of Underwater Acoustic Channel

We first analyzed the influence of underwater acoustic channels on communication signals and the performance of AMC. An example of the underwater acoustic channel with multi-path fading (Ch1) and OAN influencing a QPSK signal is illustrated in Figure 6. It is clear that underwater acoustic channel has a great influence on communication signals, especially multi-path fading.

The influence of underwater acoustic channels on the performance of AMC was analyzed through a series of contrastive experiments, in which multi-path fading effect and additive noise were considered individually. We trained an ordinary DCN without DCMF and DCCE in this section as a benchmark for subsequent experiments. Figure 7 has shown the AMC performance comparison between different channels, which include AWGN channel, OAN channel (OAN), two channels include multi-path fading and AWGN (Ch1+AWGN, Ch2+AWGN), and two channels include multi-path fading and OAN (Ch1+OAN and Ch2+OAN). We can see from Figure 7:

(1) There is an obviously positive correlation between classification accuracy and SNR.

(2) $Acc_{3}$ in AWGN channel is 14.7% higher than that in OAN channel.

(3) $Acc_{3}$ in Ch1+AWGN and Ch2+AWGN channels are 20.8% and 29.1% lower than that in AWGN channel, respectively.

Similar results can be observed when using OAN as the additive noise, in which Ch1 and Ch2 reduce the $Acc_{3}$ by 17.6% and 18.9%, respectively. This illustrates that OAN has a greater influence on the AMC performance than AWGN, especially at low SNR. The multi-path fading effect can reduce the AMC performance significantly, even at high SNR.

We also analyzed the performance of each modulation type individually in the underwater acoustic channel. Figure 8 has shown the confusion matrix charts of each modulation type in the underwater acoustic channel (Ch1+OAN). It is obvious that BPSK can be identified correctly, and the mean accuracy of PSK is higher than QAM. QPSK and 8PSK are misclassified from each other. 16QAM, 32QAM, and 64QAM are also misclassified for each other, especially 16QAM and 64QAM. The AMC accuracy improves gradually with the increase in SNR. We can conclude from the classification results that the underwater acoustic channel greatly influences the performance of AMC, especially on the high-order modulation types. The main classification errors occurred in the in-class identification of PSK and QAM.

#### 3.2.2. Comparison with Real-Valued DNN

We investigated the performance of DCN by comparing DCN with classical real-valued DNN. Two DNNs with different numbers of hidden units are built for contrastive experiments. Since DCMF and DCCE are unsuitable for real-valued DNN, an ordinary DCN without DCMF and DCCE is used for a fair comparison. The first DNN (DNN-1) is structurally similar to the DCN, and they have the same number of hidden units; the second DNN (DNN-2) is a scaled-up version of DNN-1 to keep the actual complexity and trainable parameters equal to that of the DCN. The contrastive experiments were carried out with the AWGN channel and the underwater acoustic channel (Ch1+OAN), respectively. Figure 9 has illustrated the performance comparison of DCN, DNN-1, and DNN-2 under different conditions. It can be observed that the DCN has an advantage over DNN, both in the AWGN channel and underwater acoustic channel. $Acc_{3}$ of the DCN at all SNR is higher than DNN-1 and DNN-2 at 5% and 3.4%, respectively, in the AWGN channel. In the underwater acoustic channel, DCN has a larger advantage, $Acc_{3}$ of DCN is higher than DNN-1 and DNN-2 at 8.7% and 4.3%, respectively. $Acc_{3}$ of the DCN in all conditions is higher than real-valued DNN at 5.3%. The results demonstrate that it is more appropriate to use DCN in AMC of underwater acoustic communication signals.

#### 3.2.3. Performance Analysis of Deep Complex Matched Filter

We analyzed the performance of DCMF in this section. A contrastive experiment was carried out between two DCNs with and without DCMF in the Ch1+OAN channel. In the DCN without DCMF, an ordinary CConv layer was used to ensure the parameters of the two DCNs are equal. We tracked the kernels of DCMF during the training to monitor the filtering process of the DCMF. The variations of the first kernel at different training stages are visualized in Figure 10, which includes the variation of kernel shape and spectrum. As we can see from Figure 10a that the real and imaginary parts of the complex kernel gradually approximate sine waves with the training process going. The spectrum of the complex kernel converges to a matched filter, as is shown in Figure 10b. Figure 11 has compared filtering processes between the DCMF kernel and an ordinary CConv kernel. It is obvious that the proposed DCMF can significantly remove the noise and preserve the communication signal in the received signal, while the ordinary CConv layer tends to preserve features of the full frequency band.

#### 3.2.4. Performance Analysis of Proposed Method

The AMC performance comparison between DCN with and without DCMF is shown in Figure 12. The contrastive experiments were carried out using AWGN and OAN, respectively. $Acc_{3}$ of DCN with DCMF are 1.2%, 4%, and 5.7% higher than that without DCMF in AWGN, Ch1+AWGN, and Ch2+AWGN channels, respectively. $Acc_{3}$ of DCN with DCMF are 9.5%, 4.4%, and 6.1% higher than that without DCMF in OAN, Ch1+OAN and Ch2+OAN channels, respectively. The overall $Acc_{3}$ of DCN with DCMF is 5.1% higher than that without DCMF in all conditions. It is obvious that DCMF can improve the AMC performance at low SNR(about SNR < −6 dB). The comparison results have shown the advantages of the proposed DCMF in denoising and improving the performance of AMC.

Other contrastive experiments were carried out to evaluate the performance of DCCE. The DCCE was embedded in the DCN and followed the DCMF. Multi-path fading channels Ch1 and Ch2 were considered individually. The comparison of the AMC accuracies is illustrated in Figure 13. In Figure 13a, DCCE can improve the $Acc_{3}$ in Ch1+AWGN and Ch1+OAN channels by 5.8% and 8.4%, respectively. The accuracies increments are 10.1% and 13.2% when Ch1 is replaced by Ch2, as is shown in Figure 13b. The overall $Acc_{3}$ of DCN with DCMF and DCCE is 9.3% higher than that with only DCMF in all conditions, and it is 14.3% higher than the ordinary DCN in all multi-path fading channels. The results have suggested that the proposed DCMF and DCCE can overcome the multi-path fading effect and improve the AMC performance significantly.

Another classification experiment was carried out on a real-world dataset, which was recorded in the South China Sea. The data were recorded using an omnidirectional hydrophone placed under the surface of about 10 m, the transmitter was about 3 km far away from the receiver, and the relative speed of the transmitter and receiver was less than 5 m/s. The modulation types of real-world signals include BPSK, QPSK, 16QAM, and 32QAM. The SNR of the received signals is about 3–5 dB. The received signals were down-converted and expressed as the sampled complex baseband, and all the signals were resampled and divided into 0.5 s to keep the same dimension as the simulated signals. The number of each modulation type is 100. The signals were classified by the proposed method; the classification results are shown in Table 3.

#### 3.2.5. Comparison with Achieved AMC Methods

The AMC performance comparison between the proposed method and some others achieved AMC methods were carried out. There were three AMC methods based on deep neural networks (include VGGnet [40], ResNet [41,42], LSTM [43,44]) and one AMC method using feature extraction based on HOC [45,46] (which is usually used for AMC of PSK and QAM signals) for comparison. The AMC method based on HOC used SVM as the classifier. The trainable parameters of VGGnet, ResNet, and LSTM were kept nearly to the DCN with DCMF and DCCE. The contrastive experiments were carried out with the two underwater acoustic channels (Ch1+OAN and Ch2+OAN), respectively. Table 4 and Figure 14 have illustrated the performance comparison in the two underwater acoustic channels of these mentioned AMC methods. In Figure 14a, $Acc_{3}$ of each AMC method is 50.78%, 54.94%, 55.95%, 51.48%, and 65.1%. In Figure 14b, $Acc_{3}$ of each AMC method is 59.95%, 65.79%, 65.82%, 55.17%, and 70.27%. The proposed method obtained higher classification accuracy than the other methods in the two underwater acoustic channels.

#### 3.2.6. Limitations of the Proposed Method

We carried out one experiment to analyze the SNR limitation of the proposed method. We used the combination dataset in the channels Ch1+OAN and Ch2+OAN. Figure 15 has illustrated the classification performance at lower SNR, as we can see that in multi-path fading channels when the SNR ≤−15 dB, the method can not classify the modulation types anymore.

## 4. Conclusions

In this paper, we adopted DCN into AMC of underwater acoustic communication signals, two physical signal processing layers based on DCN were proposed to improve the AMC performance. The two physical signal processing layers include DCMF and DCCE, DCMF can remove the noise from the received signals, and the DCCE can reduce the influence of multi-path fading. The proposed method can integrate the signal processing method with deep learning. Real-world underwater acoustic communication scenario was fully considered in this paper.

The results of the experiments indicated that multi-path fading and OAN in the underwater acoustic channels have a great influence on underwater acoustic communication signal and the AMC performance, especially the multi-path fading effect. DCN showed a larger advantage than real-valued DNN in AMC of underwater acoustic communication signals. By tracking and visualizing the filters of DCMF, we can observe that DCMF can significantly remove the noise in the received signals. The proposed method yielded certain performance gains and achieved robustness to the underwater acoustic channel.

## Figures and Tables

**Figure 1 entropy-25-00318-f001:**
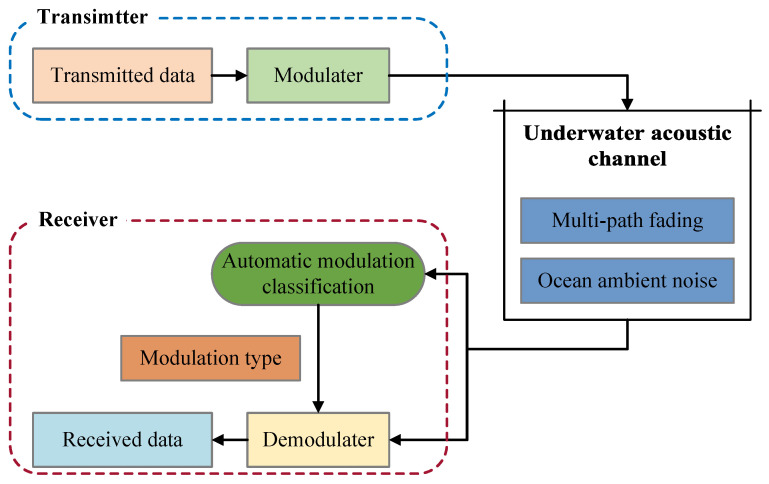
Underwater acoustic communication.

**Figure 2 entropy-25-00318-f002:**
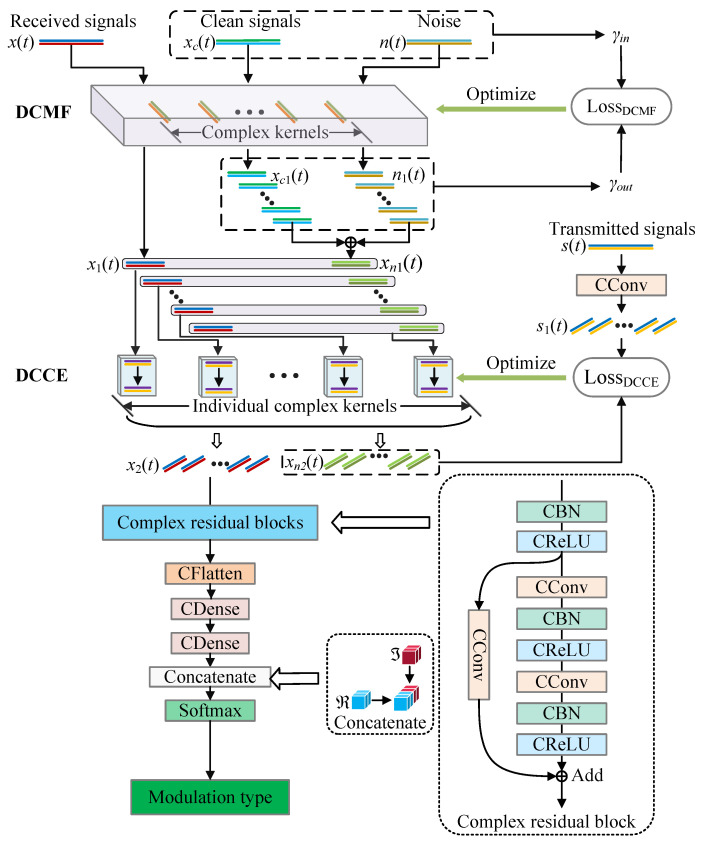
Framework of the DCN-based AMC method.

**Figure 3 entropy-25-00318-f003:**
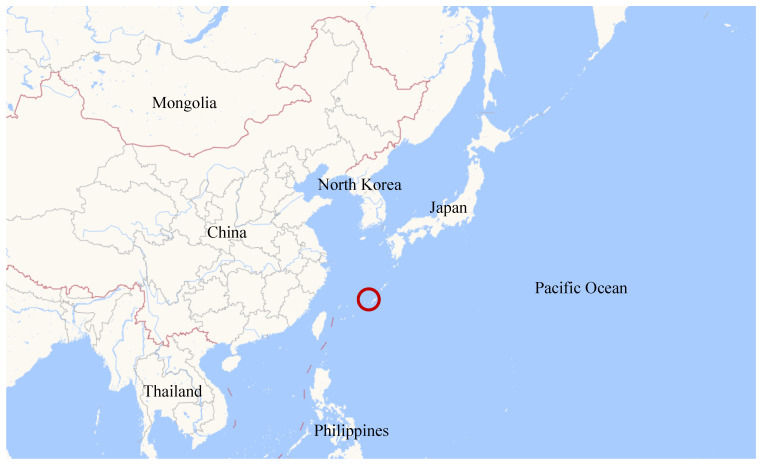
The chosen sea area in the Argo dataset.

**Figure 4 entropy-25-00318-f004:**
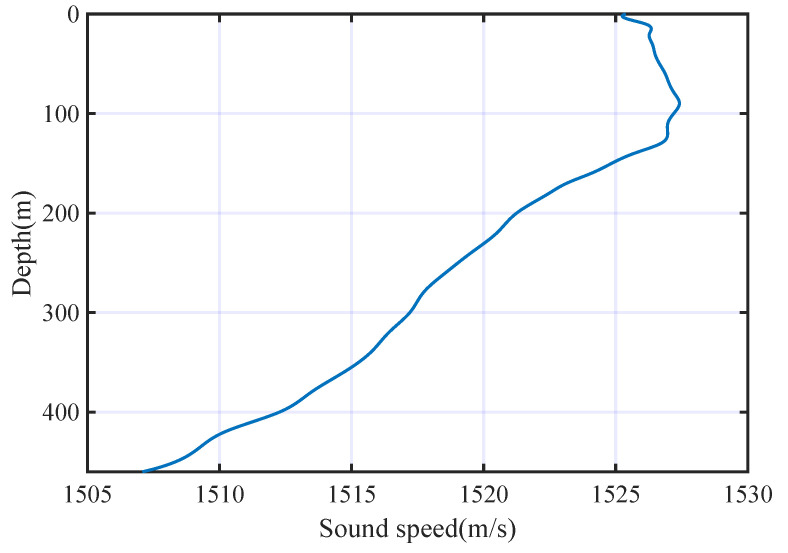
Sound velocity profile.

**Figure 5 entropy-25-00318-f005:**
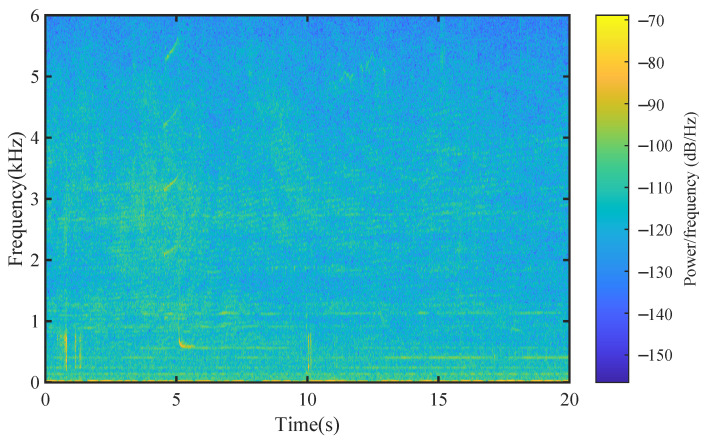
Spectrogram of one OAN sample.

**Figure 6 entropy-25-00318-f006:**
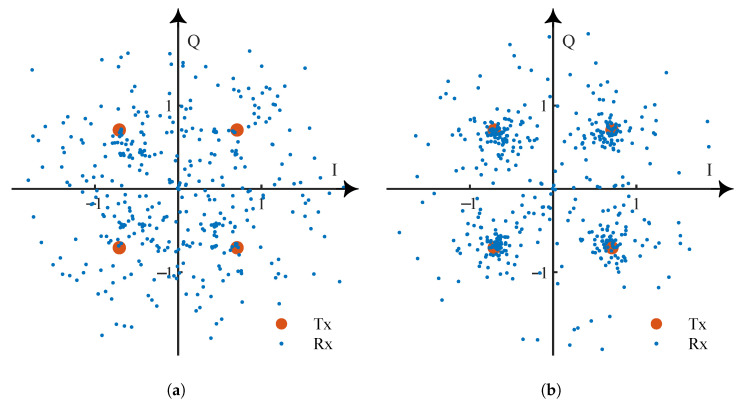
The influence of underwater acoustic channel on QPSK signal (where Tx is transmitted signal, Rx is received signal): (**a**) Influence of multi-path fading. (**b**) Influence of OAN (SNR = 0 dB).

**Figure 7 entropy-25-00318-f007:**
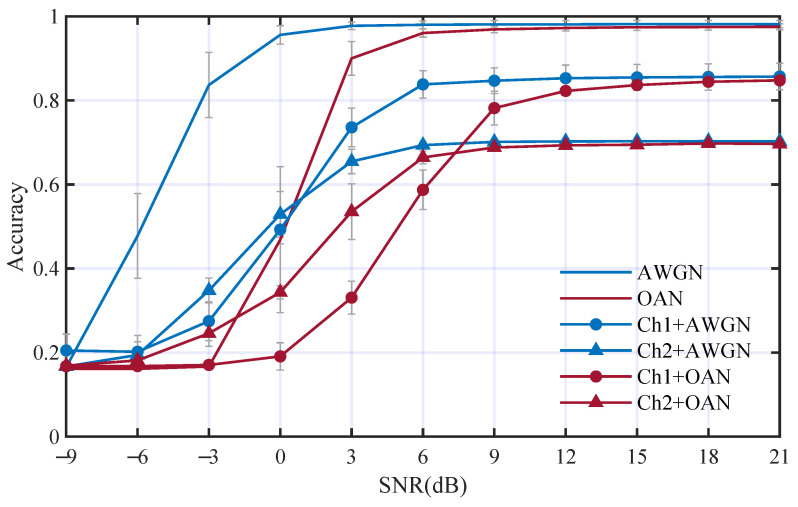
Performance comparison between different channels.

**Figure 8 entropy-25-00318-f008:**
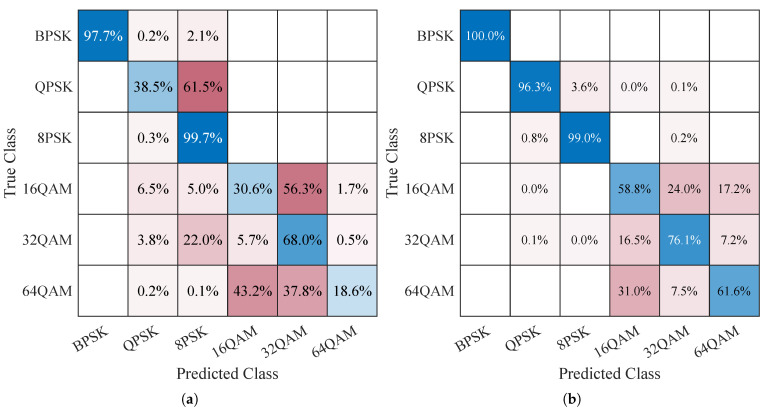
Confusion matrix charts on each modulation type in the underwater acoustic channel (Ch1+OAN): (**a**) SNR = 0 dB. (**b**) SNR = 9 dB.

**Figure 9 entropy-25-00318-f009:**
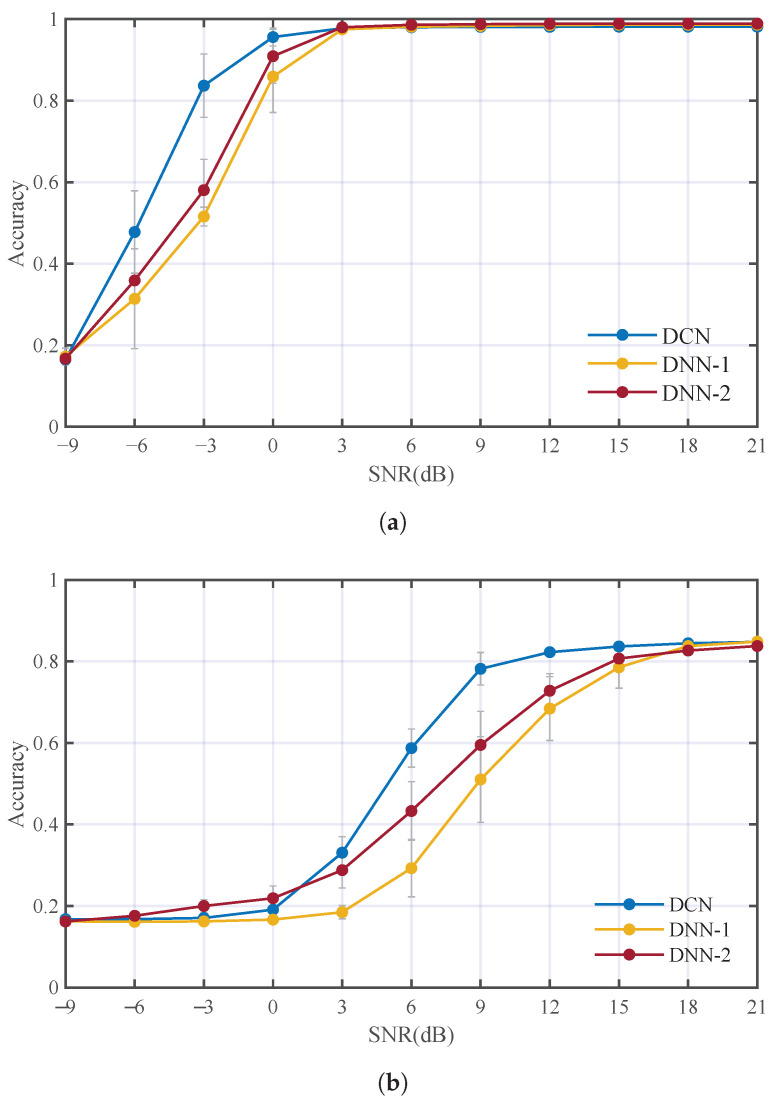
Performance comparison of DCN and real-valued DNN: (**a**) AWGN channel. (**b**) Underwater acoustic channel (Ch1+OAN).

**Figure 10 entropy-25-00318-f010:**
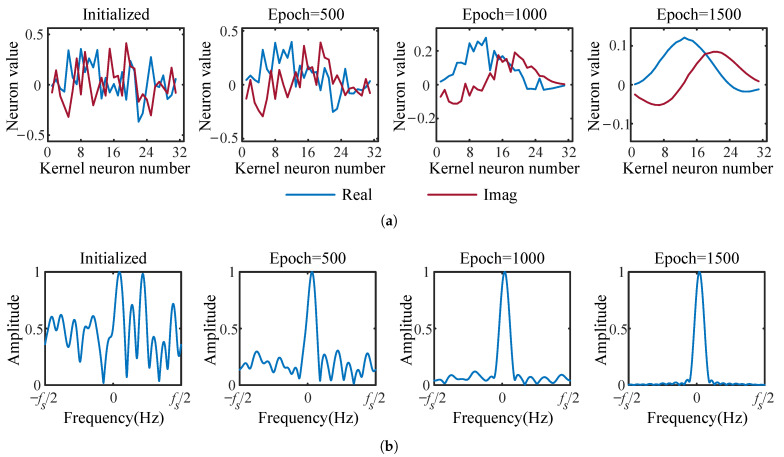
The visualization of the complex kernel of DCMF in training process (the sampling rate $f_{s}\;=\;12\;\mathrm{kHz}$, the amplitudes of spectrums are normalized to [0,1]): (**a**) Shape of the first DCMF kernel. (**b**) Spectrum of the first DCMF kernel.

**Figure 11 entropy-25-00318-f011:**
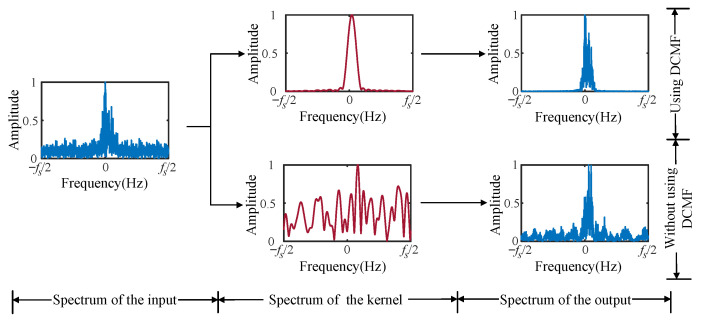
The visualization of the comparison of using DCMF and without using DCMF in frequency domain.

**Figure 12 entropy-25-00318-f012:**
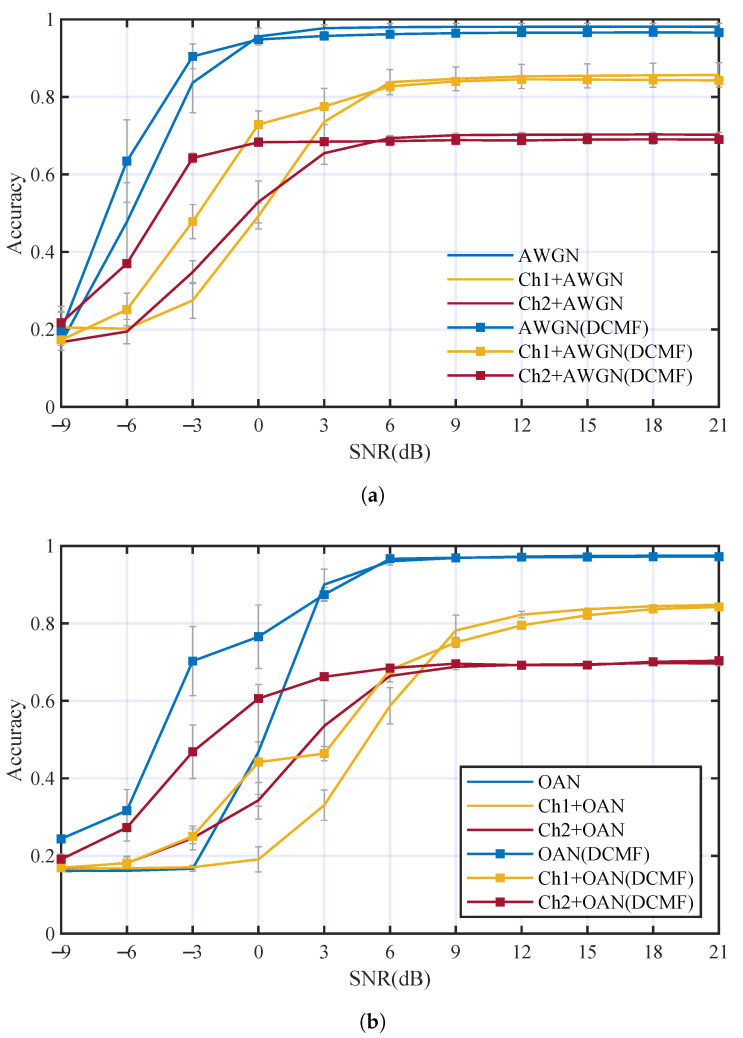
Performance comparison between DCN with and without DCMF using different kinds of noise: (**a**) AMC accuracies in different channels with AWGN, (**b**) AMC accuracies in different channels with OAN.

**Figure 13 entropy-25-00318-f013:**
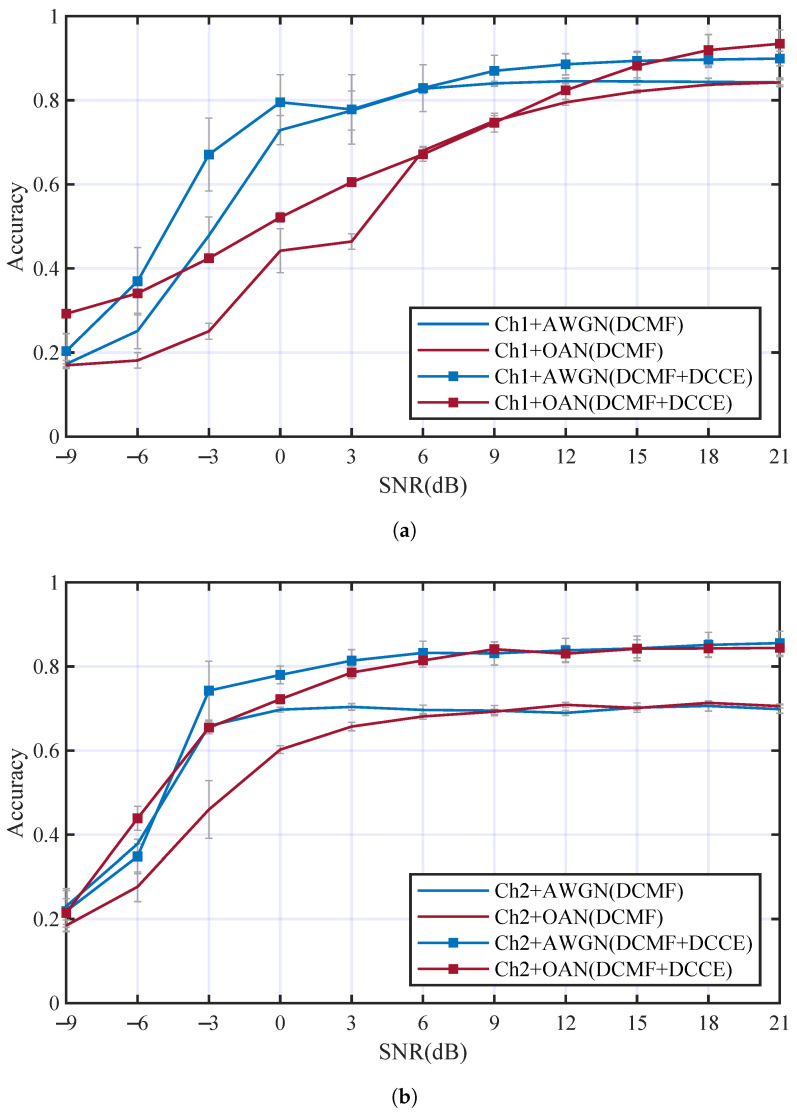
The comparison of the AMC accuracies with and without DCCE in different channels: (**a**) AMC accuracies in Ch1 with different additive noise, (**b**) AMC accuracies in Ch2 with different additive noise.

**Figure 14 entropy-25-00318-f014:**
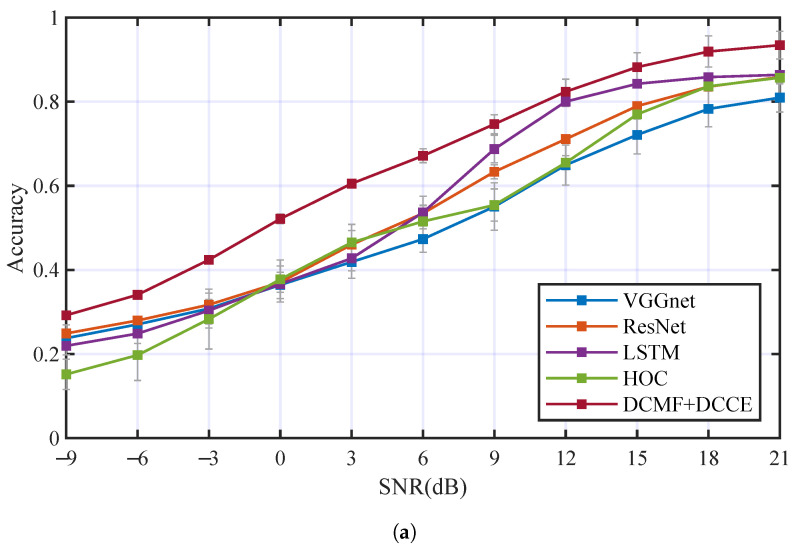
The comparison of the proposed method with others AMC methods in different channels: (**a**) AMC accuracies in Ch1+OAN, (**b**) AMC accuracies in Ch2+OAN.

**Figure 15 entropy-25-00318-f015:**
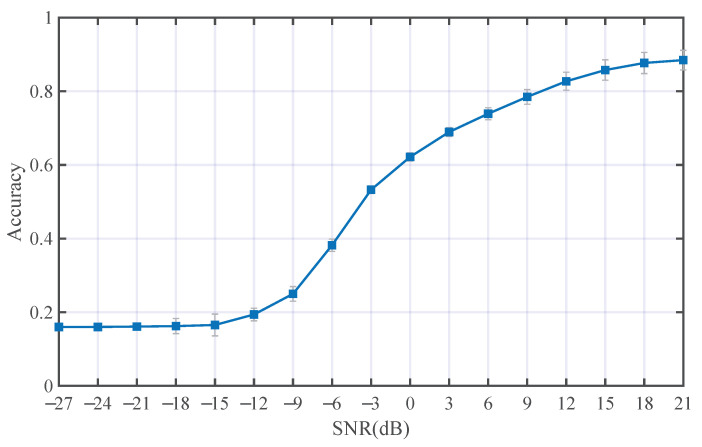
The classification result of the proposed at lower SNR (the average accuracy in Ch1+OAN and Ch2+OAN).

**Table 1 entropy-25-00318-t001:** Modulation Parameters.

Parameter	Value
Sampling rate (Hz)	12 k
Carrier frequency offset (Hz)	300
Symbol rate (Baud)	800∼1200
Roll off value	0.1∼0.4
SNR (dB)	−9∼21

**Table 2 entropy-25-00318-t002:** Time delay and amplitude of the two underwater acoustic channels.

	${\mathrm{Ch}}_{1}$		${\mathrm{Ch}}_{2}$	
Paths	TimeDelay(s)	Amplitude	TimeDelay(s)	Amplitude
1	1.965	0.861	3.276	−1
2	2.046	0.8 + 0.599j	3.334	0.305 + 0.903j
3	2.074	−0.945 − 0.153j	3.35	−0.358 − 0.848j
4	2.279	−0.143	3.357	0.483 + 0.72j
5	-	-	3.482	−0.437

**Table 3 entropy-25-00318-t003:** Classification results of the real world underwater acoustic communication signals.

	BPSK	QPSK	16QAM	32QAM
Accuracy	73%	69%	64%	71%

**Table 4 entropy-25-00318-t004:** Average accuracy at all SNR points of different methods in two underwater acoustic channels.

Channel	VGGnet	ResNet	LSTM	HOC	DCMF+DCCE
Ch1+OAN	50.78%	54.94%	55.95%	51.48%	**65.1%**
Ch2+OAN	59.95%	65.79%	65.82%	55.17%	**70.27%**

## Data Availability

The data presented in this paper are available after contacting the corresponding author.

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
