# Peer review of "Automatic Modulation Classification for Underwater Acoustic Communication Signals Based on Deep Complex Networks"

_entropy, 2023, doi:10.3390/e25020318_

Round 1
Reviewer 1 Report
The authors propose two physical signal processing layers based on DCN, which include deep complex matched filter (DCMF) and deep complex channel equalizer (DCCE). It is well structured and very interesting. However, there are some minor technical defects.
1. The author needs to further highlight the innovation of the article in the abstract. Why based on DCN? Why include DCMF and DCCE?
2. In introduction, many features have been mentioned, such as instantaneous statistics features, constellation features, spectrum features but some new nonlinear dynamic features are missing, such as 10.3390/fractalfract6100544 and 10.1016/j.chaos.2022.112400, which are more consistent with the journal “Entropy”.
3. To prove the superiority of the proposed method, it is suggested that the author increase comparative experiments from different aspects, such as data and similar methods.
4. In the conclusion, the author needs to summarize the key points and contributions of the conclusion, and some detailed descriptions about experiments and methods need to be simplified.
5. Minor grammatical errors need to be eliminated.
Reviewer 2 Report
In this paper, the authors adopted DCN into AMC of underwater acoustic communication signals. DCN has advantages over real-valued DNN for processing complex baseband signals. Since DCN is designed in the rule of complex arithmetic, it is more convenient to construct physical signal processing layer using DCN than real-valued DNN. In order to overcome the influence of underwater acoustic channel and improve the AMC performance, the authors proposed DCMF and DCCE based on DCN and embedded them in DCN to conduct a hierarchical network. Real world underwater acoustic communication scenario was fully considered in this paper. The authors simulated several underwater acoustic channels using real world ocean observation dataset and ambient noise. The paper is interesting and well-written but needs revision:
1. How was the accuracy calculated? This point is not clear.
2. The authors should present the limitations of the proposed approach.
3. The authors should present a comparison with related approaches.
4. The authors use generated signals. What is about results on real signals? This point should be presented.
Round 2
Reviewer 1 Report
The authors have solved all my concerns and the article can be published.
Reviewer 2 Report
The authors address all my concerns. The paper can be accepted.